# Towards Distributed Measurements of Electric Fields Using Optical Fibers: Proposal and Proof-Of-Concept Experiment

**DOI:** 10.3390/s20164461

**Published:** 2020-08-10

**Authors:** Regina Magalhães, João Pereira, Oleksandr Tarasenko, Sonia Martin-Lopez, Miguel González-Herráez, Walter Margulis, Hugo Fidalgo Martins

**Affiliations:** 1Departamento de Electrónica, Universidad de Alcalá, 28805 Alcalá de Henares, Madrid, Spain; sonia.martinlo@uah.es (S.M.-L.); miguel.gonzalezh@uah.es (M.G.-H.); 2RISE Research Institutes of Sweden, Electrum 236, 16440 Stockholm, Sweden; joao.pereira@ri.se (J.P.); oletar1951@gmail.com (O.T.); walter.margulis@ri.se (W.M.); 3Instituto de Óptica, Consejo Superior de Investigaciones Científicas, 28006 Madrid, Spain; hugo.martins@csic.es

**Keywords:** optical fiber sensors, Kerr effect, Pockels effect, electro-optical time domain reflectometry, optical non-linearities, distributed sensing

## Abstract

Nowadays there is an increasing demand for the cost-effective monitoring of potential threats to the integrity of high-voltage networks and electric power infrastructures. Optical fiber sensors are a particularly interesting solution for applications in these environments, due to their low cost and positive intrinsic features, including small size and weight, dielectric properties, and invulnerability to electromagnetic interference (EMI). However, due precisely to their intrinsic EMI-immune nature, the development of a distributed optical fiber sensing solution for the detection of partial discharges and external electrical fields is in principle very challenging. Here, we propose a method to exploit the third-order and second-order nonlinear effects in silica fibers, as a means to achieve highly sensitive distributed measurements of external electrical fields in real time. By monitoring the electric-field-induced variations in the refractive index using a highly sensitive Rayleigh-based CP-φOTDR scheme, we demonstrate the distributed detection of Kerr and Pockels electro-optic effects, and how those can assign a new sensing dimension to optical fibers, transducing external electric fields into visible minute disturbances in the guided light. The proposed sensing configuration, electro-optical time domain reflectometry, is validated both theoretically and experimentally, showing experimental second-order and third-order nonlinear coefficients, respectively, of *χ*^(2)^ ~ 0.27 × 10^−12^ m/V and *χ*^(3)^ ~ 2.5 × 10^−22^ m^2^/V^2^ for silica fibers.

## 1. Introduction

Electric power systems are a fundamental infrastructure of modern societies, virtually connecting homes, buildings, and cities around the world. Given the increasing complexity of electrical power networks, the timely detection of electrical faults in these systems is a general worldwide interest, since it could enable a fast response under electricity blackouts, effective electricity grid restorations, and an overall disturbance minimization [1]. Additional interests also include the prevention of catastrophic accidents when humans are involved near high-power structures [2], or even in assuring the continuous control of electrified commuter trains in large cities [3]. Although there has been an increasing investment in electric grid monitoring over the last years, the currently available data usually fails at providing operators with the necessary information to act within the necessary timeframes [4]. Issues concerning the time needed in converting data from a variety of sensors into practical information have been widely addressed, as well as difficulties in interpreting the information provided [1,5].

Nonetheless, with the increasing electricity consumption in private households worldwide, alongside with stronger environmental and regulatory restraints being enforced, the demand for highly efficient electricity grids has never been higher [6,7,8]. Dynamically controllable grids, or smart grids, presuppose the development of an empirical real-time automated technology, capable of inspecting the grid along its length and continuously report its status [9]. This type of technology is considered to be fundamental to achieve an increased reliability of the grid, and consequently improve the security and efficiency in electrical power transmission [10,11]. 

The development of this empirical distributed solution would be particularly well-suited to electricity delivery industries, envisioning the continuous monitoring of technical losses (TLs) along the network. Those can occur during the generation, transmission and distribution of electrical power, and often include power dissipation in resistive components along the grid, ground faults, or voltage leaks due to improper isolation [12]. An early TL detection by self-diagnosable grids could help to prevent extended deterioration of the electric systems, allowing for fast restoration, increased efficiency and safety, long-term production yields, and huge cost-savings in maintenance and inspection [13,14]. Furthermore, electricity providing industries could benefit greatly from the implementation of this solution, given the increasing problem of non-technical losses (NTL) faced by all electricity utilities (e.g., electricity or copper cable theft) [12,15]. Actively responsive grids could help to instantly detect and localize artificial irregularities occurring at any position in the network, preventing NTL along the system and losses of huge amounts of money by utilities [16,17].

Optical fiber sensors (OFS) are an interesting solution for application in electrical power environments, given their intrinsic features including small size and weight, dielectric nature, great flexibility, robustness, and immunity to electromagnetic interference (EMI). Over the last decades, OFS have been successfully implemented in areas such as civil engineering, security, and energy [18], providing complex structures with the ability to react under changes in the environment. More recently, with the development of distributed optical fiber sensors (DOFS), distributed alternatives became particularly attractive for large distance applications in remote locations. DFOS allow to retrieve thousands of measured points using a single optical fiber and a single interrogation unit, instead of implementing thousands of sensors and interrogation units as in the alternative case. They enable a real-time cost-effective monitoring, particularly if multiple sensors were to be installed, causing huge improvements in space and weight efficiency, overall system complexity, cost-per-sensor, and energy consumption [19]. Given these features, optical fiber-assisted electrical networks have been recently pointed out as likely solutions to revolutionize next-generation electric networks [20].

However, optical fibers are largely immune to electromagnetic disturbance, and their response to electric fields is very small, so a highly sensitive interrogation scheme is fundamental to accurately detect minute disturbances in the guided light induced by external electric fields. Recently, a Rayleigh-based technique using chirped-pulses in phase-sensitive OTDR (CP-φOTDR [21]) has been shown to provide highly sensitive distributed measurements of physical quantities such as temperature [22] and strain [23] with a simple setup, reaching a world record sensitivity of ~10^−12^ ε/$\sqrt{\mathrm{Hz}}$ for km-length distributed strain sensors over standard communication fibers [24]. CP-φOTDR has been widely demonstrated to additionally allow for continuous meter-resolved measurements of these quantities over large distances (up to 100 km [25]), provided in real time. The potential of CP-φOTDR has been increasingly recognized, being more recently proposed in areas requiring high sensitivity measurements, such as solar energy [26], oceanography, or seismology [27].

In this work, we propose a new distributed sensing technique: electro-optical time domain reflectometry. It is based on employing a CP-φOTDR to detect electro-optic effects in optical fibers, induced by external electric fields. A proof-of-concept is shown by implementing two twin-hole silica fibers filled with BiSn alloy operating as internal electrodes (separated by 40.8 μm and 25.4 μm, respectively), enabling a safe application of electric fields over several meters for the demonstration. In the first tests, we implement a twin-hole fiber which only exhibits the quadratic electro-optic effect (Kerr effect), due to the centrosymmetry properties of silica glass. In the following tests, a twin-hole fiber is implemented with an induced linear electro-optic effect (Pockels effect) achieved through a poling, exhibiting an additional second-order nonlinearity. By monitoring the electrically-induced refractive index (RI) changes in both fibers when an electrical signal is applied, we detect the Kerr and Pockels electro-optic effects in silica fibers, measuring nonlinear coefficients in relatively good agreement with the expected values. Given the minimum resolution provided by the technique, we show that the proposed distributed sensor can reach the estimated sensitivities of ~10 V/m and ~80 kV/m in quantifying external electric fields (using poled and non-poled silica fibers, respectively), provided with a few seconds of interrogation time, and over tens of thousands of individual measuring points along a single optical fiber. 

## 2. Theoretical Background

Materials change their optical properties when exposed to an electric field. This results from forces which distort the molecules constituting the material, changing their positions, shapes or orientations. An electro-optic effect is the change of the refractive index of a given material by the application of a dc or low-frequency electric field (much lower than the optical frequency). As a consequence, the refractive index of an electro-optic material is described as a function *n(E)*, where *E* is the applied electric field to the material. Since the refractive index change induced by *E* is typically very small, the variation of *n(E)* with *E* can be expanded in a Taylor’s series for *E* = 0, as follows [28]:(1)$$n(E)=n_{0}+\frac{\chi^{\left(2\right)}}{n_{0}}E+\frac{3\chi^{\left(3\right)}}{2n_{0}}E^{2}+\ldots ,$$
being *n*_0_ the refractive index when no electric field is applied to the material, *χ*^(2)^ the second-order nonlinear coefficient, and *χ*^(3)^ the third-order nonlinear coefficient. When an external electric field is applied, *χ*^(2)^ gives rise to a second order nonlinearity (SON) in the material and to the linear electro-optic (LEO) effect, which describes how the refractive index responds linearly with an applied field. The coefficient *χ*^(3)^, on the other hand, describes the third-order nonlinearity (TON) and in particular the quadratic electro-optic (QEO) effect, which represents how the refractive index changes with the square of the applied electric field. A representation of these phenomena is shown in Figure 1. 

### 2.1. Kerr Electro-Optic Effect in Silica Fibers 

Among the electro-optic materials, silica glass is particularly well-suited for application in technologies involving optoelectronics, optics or telecommunications. The transparency of silica glass together with its refractive, reflective and transmission properties allow it to be an attractive material for the manufacturing of prisms, optical lenses, optical fibers and optoelectronic materials. In addition, optical fibers made of silica glass exhibit low losses and great strength, being nowadays fundamental for application in telecommunications and optical sensing. The electro-optic response in silica glass optical fibers, however, is usually limited to the QEO, or quadratic Kerr effect, since the linear electro-optic coefficient is absent due to the inversion symmetry of fused silica. Nonetheless, the Kerr effect in silica fibers can be potentially useful for sensing applications given its typical fast response time of only a few picoseconds [29], which has been shown suitable for the detection of electrical discharges and external electric fields [30]. Additionally, with the recent development of low-cost electro-optic (EO) all-fiber devices, there has been a growing interest in exploiting the intrinsic quadratic Kerr electro-optic effect in optical fibers, namely for temperature and electric field sensing [31,32]. However, despite being the main manifestation of electro-optical response in silica glass, the QEO effect is relatively low in silica fibers (the third order nonlinear coefficient is typically *χ*^(3)^ ~ 2 × 10^−22^ m^2^/V^2^ [32]), which can be impractical for some applications.

### 2.2. Pockels Electro-Optic Effect and Poling in Silica Fibers

Although the centrosymmetry of silica glass causes the *χ*^(2)^ to be normally zero, poling has been shown to induce an effective non-zero *χ*^(2)^ coefficient in the material and the appearance of the Pockels effect. Given the increase in sensitivity provided by the additional LEO, poled fibers have been implemented in the development of electro-optic devices such as optical modulators [33], all-fiber switches, or electric field sensors [34]. This technique involves recording a permanent field in the glass that breaks its inversion symmetry, allowing for the creation of a non-zero second-order nonlinearity. In fibers, conventional poling is based on charge separation, and is carried out with the application of a high-voltage to the electrodes of the fiber and simultaneous exposure to thermal or optical excitation. When an electric field is applied to a fiber during a poling process, the resulting phase of the optical wave in a fiber is given by [35]:(2)φ(E)=2πn(E)L/λ,
where *n*(*E*) is the electric field-dependent refractive index of the material, *L* the optical path inside the medium, and *λ* corresponds to the excitation wavelength. Combining Equations (1) and (2), the phase can be expressed as:(3)$$\varphi (E)=\varphi_{0}+\frac{3\pi L\chi^{\left(3\right)}}{n_{0}\lambda }E^{2},\;with\;\;\varphi_{0}=\frac{2\pi L}{\lambda }n_{0}.$$

The relation which retrieves the induced value of *χ*^(2)^ after poling is the following [36]:(4)$$\chi^{\left(2\right)}=3\chi^{\left(3\right)}E_{rec}.$$

Therefore, after a permanent electric field $E_{rec}$ is recorded in the fiber, Equation (1) becomes:(5)$$n(E)=n_{0}+\frac{3\chi^{\left(3\right)}}{2n_{0}}{\left(E_{appl}\;+\;E_{rec}\right)}^{2},$$
being $E_{appl}$ the applied electric field. In this case, both Kerr and Pockels electro-optic effects in silica fibers are present in the material, affecting the refractive index of silica fibers under the application of an external electrical field. Combining Equation (4) into Equation (5), we finally arrive at:(6)$$n(E)=n_{0}+\frac{3\chi^{\left(3\right)}}{2n_{0}}(E_{rec}{}^{2})+\frac{3\chi^{\left(3\right)}}{2n_{0}}{\left(E_{appl}\right)}^{2}+\frac{3\chi^{\left(3\right)}}{n_{0}}(E_{rec}E_{appl}),$$
where the second term indicates a small constant increase in fiber index in the poled region, the third term describes the original quadratic Kerr effect and the last term is proportional to the applied voltage, i.e., describes the Pockels effect induced through poling. From a comparison between the last two terms, it can be noted that if the field recorded during poling $E_{rec}$ is much larger than the field to be sensed $E_{appl}$, the Pockels effect dominates over the Kerr effect.

## 3. Materials and Methods

### 3.1. Chirped-Pulse Phase-Sensitive OTDR

Chirped-pulse phase-sensitive OTDR (CP-φOTDR) is a recent advanced Rayleigh-based optical fiber reflectometry technique, proposed by Pastor-Graells et al. in 2016 [21]. The underlying principle of CP-φOTDR is based on sending chirped coherent probe pulses into the optical fiber, and recovering any temperature, RI or strain perturbations occurring locally in the fiber as local time-delays of the optical power trace. In this way, CP-φOTDR enables a truly linear response with the applied stimulus and a low variability of sensitivity along all positions of the fiber, overcoming some of the greatest limitations imposed by classic φOTDR. This technique employs a simple and robust setup with direct detection, and has been shown to enable meter-resolved distributed true measurements of temperature and strain perturbations with mK/sub-nε resolutions, provided at high interrogation speeds (kHz) and over large distances (up to 100 km) [25]. 

In this particular demonstration, a portable prototype of a CP-φOTDR interrogator was developed and installed at RISE, Sweden, enabling a sensitivity of ~10^−9^ Δn/$\sqrt{\mathrm{Hz}}$ for frequencies above a few Hz. Although a non-fully optimized CP-φOTDR was used in this case (with components of limited performance, reaching sensitivities significantly below those of state-of-art), this demonstration may be readily extended to match the limits set by conventional CP-φOTDR interrogators, which have been demonstrated to reach $∆n\sim 10^{-12}RI\sqrt{Hz}$ resolutions over sampling frequencies of 10 kHz [24].

The optical configuration of the CP-φOTDR can be seen in Figure 2, and its operation can be mainly divided in two different sections. The first part of the optical scheme (part A, red line) is responsible for generating chirped probe pulses to interrogate the optical fiber, while the other part (part B, green line) is responsible for collecting successive Rayleigh backscattered traces to detect perturbations. In this case, the pulse generation starts with a laser diode (LD) operating in continuous wave emission, controlled by temperature and current controller (I&T). The instantaneous frequency profile of the laser is modulated by directly applying a current ramp signal to the laser, created by a signal generator (SG). Finally, the laser output is time-gated by applying a square-wave signal (synchronized with the SG) to the driver of a semiconductor optical amplifier (SOA), generating linearly-chirped optical pulses. In the following steps, the pulses are respectively amplified using an erbium-doped fiber amplifier (EDFA) and controlled with a tunable attenuator to prevent non-linear effects in the optical fiber. To reduce the amplified spontaneous emission (ASE) produced by the EDFA during the amplification process, a dense wavelength-division multiplier (DWDM), used as an optical filter, is also employed in the setup. In the second part of the scheme, an optical circulator is used to send the pulses to the fiber under test (FUT), and to collect the respective reflected light. The resulting backscattered light is amplified and properly filtered before reaching the photodetector (500 MHz), being finally recorded with a high-speed (1 GS/s) digitizer. A ~1 km of standard optical fiber was placed before the FUTs in order to simulate remote operation. The electro-optical measurements were performed using 40 ns pulses linearly chirped with ~0.4GHz of total pulse spectral content, resulting in a 4-meter spatial resolution for distributed electro-optic measurements. The detection scheme was adjusted to enable a RI sampling at 1 kHz.

### 3.2. Electro-Optic Sensor with Built-In Electrodes

Since the proposed experiment targets the detection of high intensity electric fields in silica fibers across several positions, the implementation of silica fibers with built-in electrodes arose as a practical and safe solution for a proof-of-concept demonstration. Nevertheless, several parameters had to be considered when selecting the design of the envisioned electro-optic sensor, particularly to ensure a feasible and continuous interrogation with a CP-φOTDR unit. For the non-poled fiber, a solid core was required in the fiber to provide a guiding region for the CP-φOTDR, in addition to two symmetric holes to be filled with a conductive alloy material. Following these considerations, we selected the design of a twin-hole silica fiber with an outer diameter of 125 μm, whose cross-section is represented in Figure 3a (left). The geometry of the fiber encompasses a core with 8 μm of diameter, and two holes with a diameter of 28 μm each, separated by a distance of 40.8 μm (distance between internal edges). In the case of the poled fiber, the core of the fiber was needed closer to the positive electrode for thermal poling, where the depletion region is formed. Therefore, the selected design presents two holes, one closer (8.5 μm) and the other further (13.0 μm) from the core, being separated by a total distance of 25.4 μm between internal edges (see Figure 3a, at right). The holes of both fibers were filled with BiSn alloy in the molten state, which became solid at room temperature creating the built-in electrodes. To fill the holes with metal, one end of the fiber is inserted into molten metal inside a heated pressure chamber, while the other end is kept at atmospheric pressure. For the BiSn alloy used, the filling temperature was 160 °C, the pressure in the chamber was 3 atm, and the time needed to fill was less than 3 min. Longer electrodes (>3 m) can become more complex to fabricate, require higher pressure and longer time to fill. The filling technique is described in detail in [37] where is also shown the possibility to fabricate 22 m long electrodes in fiber when 8 atm was used to pressurize the chamber. Depending on the fiber used (poled or non-poled), the electrodes were ~75 cm and ~3 m long, respectively. These were electrically contacted to the exterior using thin tungsten wires, which could be connected after polishing the fiber to expose the inner electrode material. Overall, the selected design was chosen to match the features, dimensions and optical properties of standard optical fibers (SMF-28), approaching the EO fiber sensor behavior to the response of SMF without internal electrodes. Additionally, in the open-circuit system the electric current was very small, which prevents cross-sensitivities to temperature caused by internal heating during the application of an electrical field. Figure 3b represents the operation of the resulting low-loss electro-optic fiber sensor, where minute disturbances are induced in the guided light of the fiber by the application of an external electric signal in the electrodes. 

## 4. Results

### 4.1. Detection of the Electro-Optic Kerr Effect

A silica fiber filled with BiSn electrodes as the one described above was implemented to investigate the detection of the quadratic electro-optic effect in silica fibers using electro-optical time domain reflectometry. The fiber was filled with electrodes over 3 m and connected after the ~1 km of reference SMF (see Figure 2). Using a signal generator, periodic voltage sweeps between 0 V and 520 V were applied to the fiber, while monitoring the correspondent fiber RI variation. These sweeps correspond to the application of electric fields between 0 V/m and ~12.7 MV/m to the tungsten wires of the fiber, at the frequency of 50 Hz (typical transmission line frequency).

The obtained RI variation is represented in Figure 4 in (a) the time-domain and (b) the frequency-domain, showing the expected dependency: a periodic RI variation at the frequency of the applied electrical signal (50 Hz). 

The measured RI trace was averaged over 60 voltage sweeps, averaging out possible fluctuations in the electrical signal/circuit, as well as cross-sensitivities to the measurement caused by temperature drifts between measurements, or other perturbations to the fiber. However, note that averaging is not an intrinsic requirement for the proposed measurements, since the CP-φOTDR RI sensitivity can easily allow for single-shot operation [21]. The obtained results are shown in Figure 5, where the RI variation can be observed as a function of the applied electric field. 

Regarding the results in Figure 5, a parabolic fit with the form *y = a·x*^2^ was performed to the averaged signal considering the expected quadratic behavior obtained from the theory (see Equation (1), considering that *χ*^(2)^ ~ 0 in this case). From the parabolic fit, a value of *a* = 1.88 × 10^−22^ m^2^/V^2^ was obtained experimentally, which is in relatively good agreement with the reference experimental value calculated for a similar fiber (*a* = 1.48 × 10^−22^ m^2^/V^2^ [32]), measured with a different technique. The reference value of *a* was calculated using Equation (1), $a=\frac{3\chi^{\left(3\right)}}{2n_{0}}{\left(\frac{1}{1+0.18}\right)}^{2}$, considering an additional correction coefficient *η =* 0.18 for round electrodes [32], and a nonlinear coefficient of *χ*^(3)^ ~2 × 10^−22^ m^2^/V^2^ [32]. Although in this case a ~27% mismatch was obtained between the reference value and the obtained results, this is a reasonable deviation considering the conditions of the experiment. In particular, we should consider that the reference value of *χ*^(3)^ was obtained from a measurement performed to a different fiber (similar, but not the one used), and that variations in the value of *χ*^(3)^ (and therefore the value of *a*) of this order can be expected, creating a great source of error. Additionally, we should note that the length of the EO fiber used was inferior to the pulse probe width, leading to RI quantification uncertainties. Nevertheless, regardless of this discrepancy in terms of amplitude, the quadratic tendency of RI vs voltage is perfectly depicted in our measurement, which demonstrates that the CP-φOTDR is able of detecting and quantifying the quadratic electro-optic Kerr effect in silica fibers. 

### 4.2. Poled Silica Fiber: Detection of the Electro-Optic Kerr and Pockels Effect

For the second part of the experiment, the detection of the linear electro-optic effect in silica fibers was investigated by applying electric fields between 0 V/m and ~0.39 MV/m to a poled silica fiber. The fiber had been poled at 11.9 kV with two anodes at 265 °C and had a similar geometry to the previous non-poled silica fiber (see Figure 3a), presenting in this case ~75 cm long BiSn electrodes, an electrode separation of 25.4 μm, and a smaller core size (3.9 µm). This FUT was also placed after the ~1 km of SMF (simulating remote operation, see Figure 2). A signal generator was used to apply periodic voltage sweeps between 0 V and 10 V to the electrodes of the poled fiber (lower voltages required given the increased sensitivity caused by the LEO), at the frequency of 50 Hz. 

The obtained RI variation is represented in Figure 6 in (a) the time-domain and (b) the frequency-domain, showing a periodic RI variation at the frequency of the applied electrical signal (50 Hz), and the ability to detect electro-optic effects in poled silica fibers with the proposed configuration.

Similarly, as in the previous experiment, the measured RI trace was averaged over 60 voltage sweeps (over ~1 s) to mitigate experimental uncertainties and/or cross-sensitivities. The results are shown in Figure 7, where the RI variation can be observed as a function of the applied electric field. 

Since in this experiment the proposed technique allows to recover relative RI variations along the fiber instead of absolute RI values, the expected dependency of the poled fiber RI trace should follow the linear and quadratic terms of Equation (6) (note that the constant term only affects the absolute RI value). In the case of this fiber (and since *E_rec_* >> *E_appl_*), the quadratic term is practically negligible when comparing to the linear term (four orders of magnitude smaller), which means the CP-φOTDR results were expected to follow a linear tendency as represented in Figure 1b. In agreement with the theory, the obtained RI trace clearly follows a linear behavior in response to the applied electric, as it can be observed in Figure 7. Considering an approximation of Equation (6), a linear fit with the form *y = b ·x* was performed to the data shown in Figure 7. From the results, a value of *b* = 1.5792 × 10^−13^ m/V was obtained from the linear fit, which is in great agreement with the experimental value obtained for this fiber (*b* = 1.5788 × 10^−13^ m/V), measured with a Mach-Zehnder interferometer (for more information on this characterization technique see ref [39]).

The reference value of *b* was calculated using the linear term of Equation (6), $b=\chi^{\left(2\right)}\left(\frac{1}{1+0.18}\right)/n_{0}$ considering a nonlinear coefficient of *χ*^(2)^ ~ 0.272 × 10^−12^ m/V, and a correction coefficient of *η =* 0.18 for round electrodes [32].

These results demonstrate that the LEO effect can also be measured in silica fibers with electro-optical time domain reflectometry, increasing the sensitivity and potential of EO distributed fiber sensors in many energy-related applications.

## 5. Discussion

From the results, we have observed that both Kerr and Pockels electro-optic effects can be strategically used in silica fibers to achieve distributed electric field sensing based on electro-optical time domain reflectometry. Even though in this experiment two silica fibers with built-in electrodes were used for the demonstration, the achievable sensitivity in a commercial standard silica fiber without internal electrodes can be estimated, both in a poled case and in a non-poled case. Although thermal poling is a process that typically requires an optical fiber with built-in electrodes or conductive wires, those can be easily removed after poling the fiber (either by melting the electrode material, provided that the melting point of the metal used is much lower than the temperature when the cations become mobile in the fiber, as in the present case (137 °C < 265 °C), preventing the erasure of the induced Pockels effect, or simply by removing the conductive wires). Considering that optical fiber devices with internal electrodes as long as 200 m have already been fabricated [40], very long fibers can be poled to present a permanent induced Pockels effect followed by the removal of the conductive material. Therefore, if a SMF or a standard twin-hole silica fiber is placed between the plates of a capacitor (separated by a distance, *d*), both electro-optic effects can be exploited to detect the presence of an external field. For instance, given an electrode separation of 125 μm (diameter of a SMF), the expected RI changes for the calibrated fibers at 1 kV would be of Δ*n* = 9.44 × 10^−9^ for the non-poled fiber, and of Δ*n* = 1.26 × 10^−6^ for the poled fiber, values well within the CP-φOTDR demonstrated sensitivity ($∆n\sim 10^{-12}RI\sqrt{Hz}$) [24]). Furthermore, considering the CP-φOTDR sensitivity ($∆n\sim 10^{-12}RI\sqrt{Hz}$) and the features of both calibrated fibers, the sensitivity obtained with the technique is estimated to be ~80 kV/m with non-poled fibers, and ~10 V/m with poled fibers, with a few seconds of integration time. Those are compatible with the detection of partial discharges in submarine oil or electrical cables (typical transmission at ~500 kV [41]), or with the distributed monitoring of voltage leaks or electrical faults in overhead power lines, working at either low, medium or high-power transmission (usually from ~1 kV to >800 kV and spaced until up to ~1 m [42]). In the case of electric power overhead transmission, the height of the towers can range from 15 to 50 m, which could represent in a simple calculation an electric field close to 10 kV/m, at a frequency that is typically 50 Hz in Europe. The mentioned electric field is in the range of what can be measured with a poled fiber and the CP-φOTDR described. As for the transmission lines, the typical fields involved in transmission are limited to 1 kV/cm, or 10^5^ V/m, and therefore the measurement of the field with poled and unpoled fibers can be achieved with our technique. It should be noted that electric field perturbations smaller than the pulse width should be detectable with the proposed method (although probably not well quantified), as long as the average refractive index change across the resolution cell is bigger than the reported sensitivity values of the technique.

Overall, the proposed method is a safe proposal for several energy applications considering that no built-in electrodes are required, and it consists of a highly cost-effective solution providing tens of thousands of individual electro-optic sensors within a single optical fiber. Moreover, higher frequency responses could be provided when comparing with typical electric, chemical or acoustic PD detection methods [43], and in addition, since CP-φOTDR interrogators present low variability of SNR from point to point (as it has been reported in the literature [23]), the results here shown are indicative of the CP-φOTDR performance along its complete sensing range. The technique shown overcomes the intrinsic EMI immune nature of optical fibers, enabling, for the first time, a distributed solution to monitor electric fields and voltage signals in real time, with meter-spatial resolutions provided over large distances (up to 100 km [25]).

## 6. Conclusions

In this study, we have proposed a method to exploit the second and third order nonlinearities in silica fibers, as a means to achieve a real time distributed monitoring of external electric fields and stray voltages over large distances. Conceptually, the sensing method is based on measuring the minute disturbances in the guided light triggered by the application of an external electric field in the vicinity of the fiber, using a highly sensitive distributed fiber sensor based on chirped-pulse φOTDR. Given the sharp sensitivity of the interrogator in quantifying tiny variations in the refractive index (demonstrated resolutions of $∆n\sim 10^{-12}RI\sqrt{Hz}$ [24]), this method has been shown capable to enable an accurate detection of the electrically induced Kerr and Pockels electro-optic effects in silica fibers, overcoming the difficulties imposed by their EMI immunity intrinsic nature. A proof-of-concept was shown by applying electrical fields to poled and non-poled twin-hole silica fibers filled with BiSn alloy electrodes, in which both Kerr or Pockels electro-optic effects were effectively detected. An experimental third-order nonlinear coefficient of *χ*^(3)^ ~ 2.5 × 10^−22^ m^2^/V^2^ was obtained from the results with a non-poled silica fiber, while a second-order nonlinear coefficient of *χ*^(2)^ ~ 0.27 × 10^−12^ m/V was measured from the results with a poled silica fiber. These values are in relatively good agreement with the reference values from similar fibers measured with different techniques (*χ*^(3)^ ~ 2.0 × 10^−22^ m^2^/V^2^ and *χ*^(2)^ ~ 0.27 × 10^−12^ m/V, respectively), demonstrating the ability to detect electro-optic effects with the proposed technique.

This sensing configuration has a high potential for integration in energy industries given the attainable sensitivity of ~80 kV/m and ~10 V/m for non-poled and poled fibers, respectively, using an optimized state-of-the-art CP-φOTDR [24]. Since this method turns electric signals into measurable perturbations in the fiber core along its length, this distributed sensor allows for tens of thousands of individual electro-optic sensors along a single optical fiber, each one capable of reporting the status of the electrical environment in real time. The proposed sensor can be implemented to monitor TL and NTL in very large and complex networks, or to detect electrical flaws in submarine cables, large capacitors or other high voltage infrastructures, allowing overall for an increased efficiency, optimized management, and cost reduction in maintenance and inspection. Additionally, since it allows for meter-resolved measurements in real time over large distances (up to 100 km [25]), the proposed distributed sensor can be an overall important step towards the development of smart grids, leading to a new generation of reliable, safe, efficient and sustainable electrical systems.

## Figures and Tables

**Figure 1 sensors-20-04461-f001:**
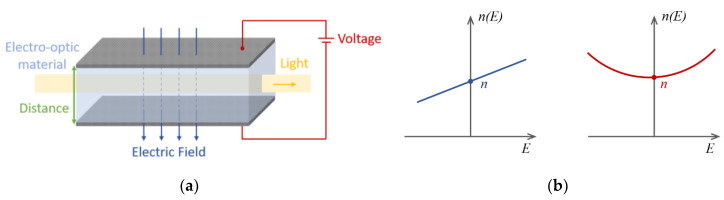
(**a**) The application of a steady electric field to an electro-optic material results in the change of its refractive index, modifying the effect of the material over traveling light.; (**b**) Representation of the LEO and QEO, respectively, in an electro-optic material.

**Figure 2 sensors-20-04461-f002:**
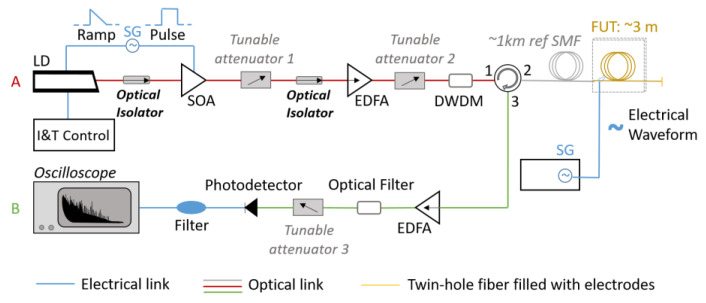
Optical configuration of a chirped-pulse phase-sensitive OTDR (CP-φOTDR): acronyms are explained within the text.

**Figure 3 sensors-20-04461-f003:**
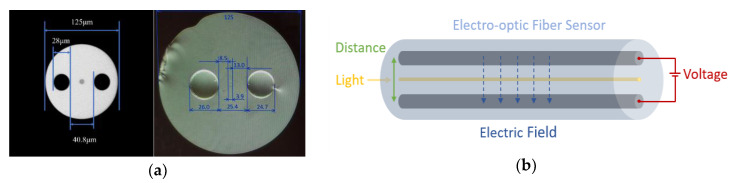
(**a**) Cross-section of the proposed twin-hole silica fiber (left) [38], reprinted with permission from [35] ^©^ The Optical Society, and of the thermally poled fiber (right). (**b**) The application of a dc electric field to the electrodes of the electro-optic sensor results in the change of the refractive index in the core of the fiber.

**Figure 4 sensors-20-04461-f004:**
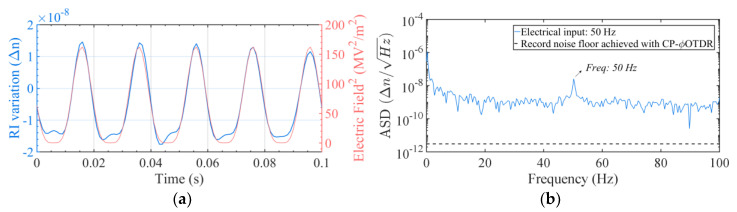
(**a**) Measured RI variation for a periodic electric field applied to a non-poled silica fiber at 50 Hz, represented in the time-domain (blue line); vs the applied signal (proportional to E^2^, varying between 0 V^2^/m^2^ and ~161 MV^2^/m^2^) applied to the fiber (red line). (**b**) Amplitude Spectral Density from the obtained RI signal (blue line). The record noise floor achieved with in CP-φOTDR in standard SMF is also represented (black dashed line) to contextualize the full potential of the technique [24], although it is not aimed at a direct comparison with the presented results.

**Figure 5 sensors-20-04461-f005:**
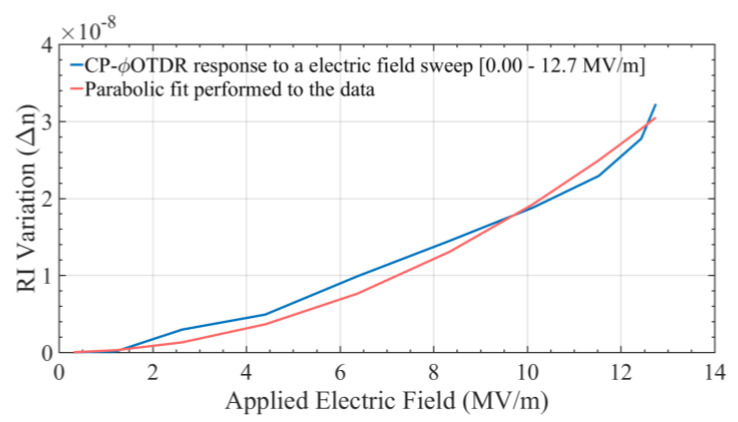
Blue line: RI variation measured by the EO sensor, averaged over ~1 s (60 voltage sweeps). Red line: parabolic fit performed to the data.

**Figure 6 sensors-20-04461-f006:**
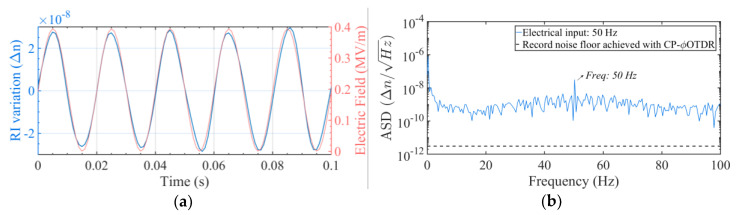
(**a**) Measured RI variation for a periodic electric field applied to a poled fiber (50 Hz, between 0 V/m and ~0.39 MV/m), via the built-in electrodes, represented in the time-domain (blue line); expected variation (proportional to applied E; see discussion below) is also presented for comparison (red line). (**b**) Measured RI variation represented in the frequency domain (blue line). The record noise floor achieved with CP-φOTDR in standard SMF is also represented [black dashed line in (**b**)] to contextualize the full potential of the technique [24], although it is not aimed at a direct comparison with the presented results.

**Figure 7 sensors-20-04461-f007:**
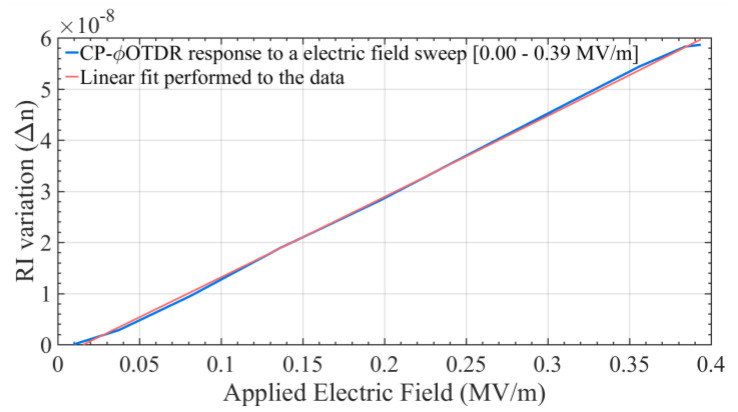
Blue line: RI variation measured by the EO sensor. Signal is averaged over ~1 s (equivalent time for each period). Red line: linear fit performed to the data.

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
