# Peer review of "Towards Distributed Measurements of Electric Fields Using Optical Fibers: Proposal and Proof-Of-Concept Experiment"

_sensors, 2020, doi:10.3390/s20164461_

Round 1

Reviewer 1 Report

This article describes a proposed distributed fiber optic electric field sensor for use monitoring high voltage networks and electric power infrastructure. There has been very little work investigating fiber optic sensors for electric field monitoring and recent advances in DAS make this topic worth re-investigating. As a result, I think this can be a nice contribution. In terms of experimentation, the authors characterize the Kerr effect in un-poled fiber and the Pockels effect in a poled fiber using a CP-phi-OTDR system and project the electric field sensitivity they would expect in a distributed system.

Since this article is primarily a proposal, I would have liked to see more discussion of the application requirements. Particularly since electrical infrastructure monitoring is not a familiar field for most readers familiar with fiber optic sensing. For example, the authors discuss the performance limits of CP-phi-OTDR sensors, but the range, spatial resolution, and sensor bandwidth requirements for high voltage sensing applications were not really discussed.

In the discussion section, the authors estimate that the proposed system would be sensitive to either 80 kV/m (in the non-poled case) or 10 V/m and then claim that “those are compatible with the detection of partial discharges in submarine oil or electrical cables (typical transmission at 500 kV)”. How does the transmission voltage of 500 kV translate to the required sensitivity in units of V/m? The poled and non-poled fiber sensitivities are 3 orders of magnitude different—are either of them acceptable for the target application? Or is the non-poled fiber only useful for a subset of the potential applications?

I also had some questions about the experimental demonstration.

As I understand it, this approach assumes the electric discharge you want to detect will produce a uniform electric field across meters of fiber. This is ensured in the demonstration by including electrodes that run parallel to the optical core, but is this a fair assumption in the proposed use case?

Why were different fibers used in the poled and non-poled experiments? 

The electrodes in both cases are smaller than the 4 m spatial resolution and in the poled case, the electrodes only covered 75 cm. Can the authors discuss this choice and how this mismatch was accounted for when extracting the non-linear coefficients?

In Fig. 4, the non-poled sensor should provide a quadratic response to the applied electric field. However, in Fig. 4(a), the response appears to be pretty linear, effectively mirroring the shape of the applied electric field. The dependence in Fig. 5 is fit as a quadratic, but it looks like a linear fit would provide nearly as good a match to the data. Can the authors comment on this?

In section 3.1, the authors mention that they are using a portable CP-phi-OTDR system and that the strain sensitivity is 3 orders of magnitude worse than previously demonstrated. This is a pretty huge difference in performance – can the authors elaborate on what components are used here that are so much worse than in previous demonstrations?

On page 9, the authors compare the measured linear electro-optic coefficient with the value obtained in Ref. 37. Saying the measured value “is in great agreement with the reference experimental value obtained for this fiber measured with a different technique [37].” But as far as I can tell, Ref. 37 used a very different fiber (e.g. it included two cores), so I’m not sure how to interpret this comparison.

Reviewer 2 Report

In this work the authors present an interesting fiber sensor design for detecting the variation of external electric field (in particular for high power application) by using a silica fiber, exploiting the non-linear refractive index change, and reading the variation using a CP phi-OTDR technique. The fibers, used for the experimental demonstration, are the typical twin holes fibers with holes filled with metallic electrodes. One experiment concentrates to demonstrate the device by using the intrinsic Kerr effect of the silica, while the second exploits the Pockels effect induced by a poling process. The authors claim that this kind of sensor can be used for distributing measurement over long distance.

The work is in general well written and well organized. The topic is also of great interest. Nevertheless, I have some concerns that must be addressed before to accept the work. In particular I found that the experimental work is incomplete to foster the claim of the authors:

1) The authors filled the holes with a metal alloy. Can the authors comment on the procedure of filling the holes? What is the temperature of the melted alloy? Is it difficult to fill 3 meters of fiber? Is it necessary a special equipment to do so?

2) Alternatively, the authors said that the holes can be filled with filament electrodes. Is it really possible to fit 3 meters of metallic filament thinner than 28 um inside the fiber? the author should discuss. 

3) Just out of curiosity, why the Dn in Fig. 4(a) is negative?

4) The authors claim that the same results can be achieved by using a fiber with no electrodes or a fiber where the electrodes have been removed. I can totally agree in the case of Kerr effect, I have some doubts regarding the case of poled fibers showing Pockels effect. If Pockels effect has been induced by thermal poling, can the thermal treatment, used to remove the electrodes, totally or partially restore the centro-simmetry, thus vanishing the Pockels effect? The authors must discuss on this. The best would be to organize an experiment where the electrodes are removed and the electric field is induced by external plates connected to a generator.

5) The authors claim that their sensor system permits distributed measurement. In theory this is true, but no result has been shown to support this claim. How can I know that the system reveals the field variation at 1 km of distance from the detector? If I cut the SMF pigtail fiber to 500 meters, can the system reveal that the field is applied at 500 meters from the detector. These results should be shown and discussed. In my opinion (maybe I am asking to much) the best would be to organize an experiment, where two sensors (possibly with the electrodes removed) are splices in two different positions of the output line, and show that the system is able to detect the field from both sensors.

Round 2

Reviewer 2 Report

The authors did a good job to reply to my previous comments and the work significantly improved. Moreover, I found convincing the motivation regarding the convenience to perform or not an experimental demonstration of the distributed measurement in this phase of the research. It remains the fact that I strongly suggest that kind experiment for the future development of this research field.

For now, I am fine with the current version of the paper that can be accepted.